# A Prospective Analysis of the Effects of a Powder-Type Hemostatic Agent on the Short-Term Outcomes after Liver Resection

**DOI:** 10.3390/medicina60020278

**Published:** 2024-02-05

**Authors:** MeeYoung Kang, Jai Young Cho, Ho-Seong Han, Yoo-Seok Yoon, Hae Won Lee, Boram Lee, Yeshong Park, Jinju Kim

**Affiliations:** Department of Surgery, Seoul National University Bundang Hospital, Seoul National University College of Medicine, Seoul 13620, Republic of Korea

**Keywords:** liver resection, topical hemostatic agents, microporous polysaccharide hemosphere

## Abstract

*Background and Objectives*: Postoperative bleeding is a significant cause of morbidity and mortality following liver resection. Therefore, it is crucial to minimize bleeding during liver resection and effectively manage it when it occurs. Arista^®^ AH (Becton, Dickinson and Company, Franklin Lakes, NJ, USA) is a microporous polysaccharide hemosphere (MPH), a new plant-derived polysaccharide powder hemostat that can be applied to the entire surgical field. This study prospectively assessed the effectiveness of Arista for bleeding control when applied intraoperatively to the liver resection surface. *Materials and Methods*: Data were collected at Seoul National University Bundang Hospital for patients who underwent liver resection owing to malignant hepatocellular carcinoma or benign liver diseases. We compared the outcomes between 45 patients managed with Arista^®^ AH (data were prospectively collected between September 2022 and May 2023) and 156 patients managed without the use of Arista^®^ AH (data were retrospectively collected between January 2021 and December 2021). *Results*: There were no significant differences in patient characteristics between the two groups. The estimated blood loss (EBL) was significantly lower in the Arista^®^ AH group compared with the control group (495.56 ± 672.7 mL vs. 691.9 ± 777.5 mL, *p* = 0.049). The mean postoperative hospital stay was significantly shorter in the Arista^®^ AH group (5.93 ± 1.88 days vs. 6.94 ± 4.17 days, *p* = 0.024). The time to Jackson-Pratt drain removal was also significantly shorter in the Arista^®^ AH group (4.64 ± 1.31 days vs. 5.30 ± 2.87 days, *p* = 0.030). The patient subgroup was divided into four categories based on the type of resection and the presence or absence of cirrhosis. Within the subgroup of major resections in non-cirrhotic patients, the Arista^®^ AH group demonstrated significantly better outcomes compared to the control group, showed lower EBL, reduced need for blood transfusions, decreased volume of drain fluid collected within 48 h, earlier removal of drains, and shorter hospital stays. In contrast, for the other subgroups such as minor resection (both non-cirrhotic and cirrhotic) and major resection with cirrhosis, the differences between the Arista^®^ AH and control groups in various parameters like EBL, blood transfusion rates, drain fluid volume, time to drain removal, and duration of hospital stay were not statistically significant. *Conclusions*: Arista^®^ AH significantly improved intraoperative blood management and postoperative recovery in patients undergoing liver resection, particularly in non-cirrhotic patients who underwent major resection.

## 1. Introduction

Postoperative bleeding is a significant cause of morbidity and mortality following liver resection. Perioperative bleeding and blood transfusion significantly increase the rates of mortality and major morbidity and are responsible for longer hospital stays [1]. Therefore, it is crucial to minimize bleeding during liver resection and to effectively manage it when it occurs. Despite the use of meticulous surgical techniques and advanced equipment, blood may ooze from the transected raw liver surface, especially when liver resection leaves a deep empty cavity.

Technical advancements in hemostatic agents have enhanced the safety of hepatic surgery [2,3]. Topical hemostatic agents (THAs) are a group of synthetic and biological products, such as collagens, fibrins, and cyanoacrylates, that are designed to facilitate hemostasis through vessel sealing techniques. Previous studies have revealed that using THAs in liver resection effectively reduces the time to hemostasis (TTH) and minimizes the perioperative transfusion rate [4,5,6]. THAs have undergone significant transformations, increasingly focusing on biocompatibility and efficiency.

Arista^®^ AH (Becton, Dickinson and Company, Franklin Lakes, NJ, USA) is a 100% plant-based absorbable surgical hemostatic powder derived from purified plant starch. The effects of Arista^®^ AH are achieved via its formulation as microporous polysaccharide hemosphere (MPH), a patented blood clotting technology that received premarket FDA approval in 2006. It is indicated in various surgical procedures (except neurological, ophthalmic, and urological) as an adjunctive hemostatic device. Arista^®^ AH is utilized when control of capillary, venous, and arteriolar bleeding through pressure, ligature, and other conventional methods is deemed ineffective or impractical. This approval highlights its broad applicability and effectiveness in managing bleeding in diverse surgical contexts. Arista^®^ AH is simple and safe to use. It requires no mixing and no refrigeration, and the powder is applied directly to the bleeding site. It is known to be safe because it is free of thrombin, biocompatible, nonpyrogenic, and typically absorbed within 24–48 h by the activity of amylases. According to the Arista^®^ AH PMA P050038 clinical study, the clotting process begins as soon as Arista^®^ AH powder is applied to the surgical field, regardless of the patient’s coagulation status [7]. In a porcine punch liver biopsy model, Arista^®^ AH demonstrated superior hemostatic effectiveness compared to a commercially available porcine gelatin sponge; it achieved complete hemostasis in 89% of treated sites within 5 min and 100% within 10 min, with an average time to hemostasis of 155 ± 112 s, significantly faster than the 322 ± 137 s for the gelatin sponge. Arista^®^ AH represents a significant advancement in hemostatic technology, offering unique benefits with its microporous structure and ease of application. These features expedite hemostasis and potentially reduce the need for blood transfusions, contributing to improved surgical outcomes and shorter hospital stays. Additionally, Arista^®^ AH absorbs water and low-molecular-weight compounds from the blood, concentrating platelets and clotting proteins at its beaded surface, thereby enhancing endogenous clotting processes. Previous studies have indicated that, in surgeries other than liver resection, Arista^®^ AH reduced the TTH and the postoperative blood transfusion requirement [7,8]. However, no studies have specifically investigated whether Arista^®^ AH minimizes or effectively manages bleeding during liver resection.

Therefore, this study sought to investigate the efficacy of Arista^®^ AH for controlling bleeding when applied intraoperatively to the raw liver resection surface.

## 2. Materials and Methods

This study was approved by the Ethics Committee of Seoul National University Bundang Hospital (approval number: B-2208-773-303, date of approval 26 July 2022) and conducted in accordance with the Declaration of Helsinki.

### 2.1. Inclusion Criteria

Data were prospectively collected at Seoul National University Bundang Hospital for patients who underwent liver resection owing to malignant hepatocellular carcinoma or benign liver diseases and satisfied the following criteria: (A) patients scheduled for open or minimally invasive liver resection; (B) anatomical or non-anatomical liver resection; (C) age ≥ 19 to ≤80 years; (D) male or female; (E) American Society of Anesthesiologists physical status I–IV; (F) body mass index BMI ≤ 40 kg/m^2^; and (G) no concurrent resection of other organs, such as the bile duct, colon, or duodenum. We compared 45 patients who underwent liver resection with the use of Arista^®^ AH (between September 2022 and May 2023) to 156 patients who underwent liver resection with conventional hemostatic agents (between January 2021 and December 2021). In this study, prior to the initiation of Arista^®^ AH use in September 2022, different types of THA were used in liver resection. Consequently, patients who underwent liver resection in 2021 were selected as the control group. Patients undergoing liver resection were classified into four subgroups to assess the impact of hepatic status on bleeding and coagulation risks. This classification was based on the extent of liver resection (major or minor resection) and the presence or absence of cirrhosis. All patients underwent the necessary preoperative assessments, including overall assessment, liver function, tumor markers, hepatitis infection status, and imaging examinations, such as spiral computed tomography or magnetic resonance imaging.

### 2.2. Surgical Technique and Outcome Indicators

There were no major changes in the surgical techniques or protocol during the study period. Hepatic parenchymal transection was performed using a technique combined with an ultrasonic surgical aspirator CUSA^®^ (Integra, Inc., Charlotte, NC, USA). During resection, bleeding was controlled using conventional methods, such as suture, ligation, or monopolar/bipolar hemostasis. After liver resection, bipolar hemostasis was initially conducted, followed by a thorough visual inspection to confirm the absence of any manageable bleeding. Once the absence of bleeding was checked, topical hemostatic agents were applied to the resection site. The control group received at least one hemostatic agent such as Tacosil^®^ (Takeda Pharmaceuticals International GmbH, Zurich, Switzerland), FloSeal^®^ (Baxter International, Inc., Deerfield, IL, USA), or Surgicel^®^ (Ethicon, Inc., Bridgewater, NJ, USA) without Arista^®^ AH. Among the Arista^®^ AH group patients, Arista^®^ AH (5-*g* size) was applied to the resected surface of the liver following primary active bleeding control. The number of Arista^®^ AH (5-*g* size) applications was dependent upon bleeding and at the discretion of the practicing surgeon. A Jackson-Pratt (JP) drain was inserted at the resection site in all patients.

The primary outcomes were intraoperative bleeding (estimated blood loss (EBL)), red blood cell transfusion amount, the volume of drain fluid collected in 48 h after surgery, and the time between surgery and drain removal. The secondary outcomes were the incidence of complications within 30 days of operation (graded according to the Clavien-Dindo classification) and the duration of the postoperative hospital stay.

### 2.3. Statistical Analyses

All data were analyzed using SPSS statistical package version 27.0 (SPSS Inc., Chicago, IL, USA). The two groups were compared using the Student’s *t* test for continuous data and the χ^2^ test or Fisher’s exact test for categorical data. All data are expressed as the mean ± standard deviation or as the median and range. *p*-values of ≤0.05 were considered significant.

## 3. Results

### 3.1. Patient Characteristics and Subgroup Analysis

A total of 201 patients were recruited for this study, with 45 patients in the Arista^®^ AH group and 156 in the non-Arista^®^ AH control group. Table 1 shows the baseline characteristics of both groups. There were no significant differences in patient characteristics between the two groups. The gender distribution was 28.9% female and 71.1% male in the Arista^®^ AH group, compared to 24.4% female and 75.6% male in the control group. The median age was slightly lower in the Arista^®^ AH group (55.8 years) than in the control group (60.24 years), but this difference was not statistically significant. Similarly, the prevalence of underlying liver diseases such as HBV, HCV, alcohol-related liver disease, and non-B/non-C hepatocellular carcinoma was comparable. Body mass index (BMI) averages and tumor sizes were also closely matched between the groups. Furthermore, the proportion of patients with cirrhosis and the distribution of major versus minor liver resections were similar, indicating a consistent baseline across the study cohorts for a reliable comparison of the outcomes related to the use of Arista^®^ AH versus the control group. Liver function tests and creatinine levels were comparable between the Arista^®^ AH group and the control group. Both groups showed similar average levels of aspartate aminotransferase (AST) and alanine aminotransferase (ALT), with no significant differences observed. Additionally, the prothrombin time–international normalized ratio (PT INR) values, which indicate blood clotting ability, were also equivalent between the two groups. The creatinine levels, indicative of kidney function, were consistent and within normal ranges in both groups. These results suggest that liver and kidney functions were not significantly impacted by the choice of hemostatic agent. Patients were also divided into four subgroups according to the extent of surgery (major or minor resection) and the presence or absence of cirrhosis (Table 2). The minor resection, non-cirrhotic subgroup comprised 67 patients in the control group (42.9%) and 15 in the Arista group (33.3%). The minor resection cirrhotic subgroup comprised 43 patients in the control group (27.6%) and 15 in the Arista group (33.3%). The major resection non-cirrhotic subgroup comprised 25 patients in the control group (16%) and 12 in the Arista group (26.7%). The major resection cirrhotic subgroup comprised 21 patients in the control group (13.5%) and 3 in the Arista group (6.7%).

### 3.2. Outcomes

The mean operation time tended to be shorter in the Arista^®^ AH group, with a mean ± SD of 155.56 ± 67.0 min, versus 179.01 ± 124.5 min in the control group, although this difference was not statistically significant (*p* = 0.099) (Table 3). The EBL was significantly different between the two groups, being significantly less in the Arista^®^ AH group (495.56 ± 672.7 mL vs. 691.9 ± 777.5 mL, *p* = 0.049). The percentage of patients requiring blood transfusion was similar in both groups, being required in 13.3% (6 patients) of patients in the Arista^®^ AH group versus 19.2% (30 patients) in the control group (*p* = 0.508). The volume of drain fluid collected in 48 h after surgery was significantly lower in the Arista^®^ AH group than in the control group (337.13 ± 273.8 mL vs. 441.13 ± 388.3 mL; *p* = 0.045). Moreover, the time from surgery to drain removal was significantly shorter in the Arista^®^ AH group (4.64 ± 1.31 days vs. 5.30 ± 2.87 days; *p* = 0.030). In addition, the duration of hospital stay was significantly shorter in the Arista^®^ AH group (5.93 ± 1.88 days vs. 6.94 ± 4.17 days; *p* = 0.024).

The overall complication rate in the Arista^®^ AH group was 22.24% (10 patients) compared with 25% in the control group (39 patients), which was not significantly different (*p* = 0.853). Bleeding-related complications occurred in 8.9% of patients in the Arista^®^ AH group (4 patients) versus 7.7% in the control group (12 patients). Bile leakage was observed in 1.1% of patients in the Arista^®^ AH group (one patient) versus 2.6% in the control group (four patients). Clinically relevant complications (Clavien-Dindo classification ≥ IIIa) were reported in 6.7% of patients in the Arista^®^ AH group (three patients) versus 7.7% of patients in the control group (12 patients) (*p* = 0.888), indicating the frequency of clinically relevant complications was not significantly different between the groups (Table 3).

### 3.3. Subgroup Analyses Based on Resection Type and Cirrhosis Status

In the subgroup analysis, when patients were divided into four groups by resection type and cirrhosis status, significant differences were observed between the Arista^®^ AH and control groups in the major resection, non-cirrhotic subgroup (Table 4). In this subgroup, the EBL was significantly lower in the Arista^®^ AH group than in the control group (780 ± 400.8 mL vs. 1076 ± 777.5 mL; *p* = 0.036). The percentage of patients requiring blood transfusion was also lower in the Arista^®^ AH group (8.3%) than in the control group (36.0%), although this was not statistically significant (*p* = 0.076). The volume of drain fluid collected in 48 h was significantly less in the Arista^®^ AH group than in the control group (556.1 ± 288.7 mL vs. 735.4 ± 333.5 mL; *p* = 0.049). Furthermore, the time from surgery to drain removal was shorter in the Arista^®^ AH group than in the control group (5.08 ± 0.8 days vs. 6.24 ± 2.04 days; *p* = 0.019). In addition, the duration of hospital stay was shorter in the Arista^®^ AH group (6.33 ± 0.98 days vs. 7.56 ± 2.13 days; *p* = 0.013). In contrast, for the other subgroups such as minor resection (both non-cirrhotic and cirrhotic) and major resection with cirrhosis, the differences between the Arista^®^ AH and control groups in various parameters like EBL, blood transfusion rates, drain fluid volume, time to drain removal, and duration of hospital stay were not statistically significant. These findings indicate that while Arista^®^ AH shows considerable effectiveness in major non-cirrhotic liver resections, its impact in other types of resections, particularly those involving minor resections or patients with cirrhosis, is less pronounced or statistically non-significant based on the parameters measured in this study.

## 4. Discussion

This study was designed to evaluate the effectiveness of Arista^®^ AH for managing bleeding during liver resection. Our primary focus was to determine perioperative outcomes by assessing how Arista^®^ AH controls bleeding when applied directly to the liver surface during surgery. The results show that Arista^®^ AH significantly improved the perioperative outcomes after liver resection. In particular, the Arista group showed a reduction in EBL compared with the control group, demonstrating its effectiveness in managing intraoperative bleeding. In addition, patients in the Arista^®^ AH group had a shorter postoperative hospital stay, and the JP drain was removed sooner, indicating an accelerated recovery. These results were particularly evident in patients undergoing major resections without cirrhosis.

According to recent studies examining the trends in liver cancer in Korea over the past 20 years, it is evident that increased early detection of hepatocellular carcinoma has led to an improvement in the 5-year survival rate among patients undergoing surgical treatment [9,10,11]. Surgeons utilize numerous techniques to reduce blood loss during partial liver resection. Several studies have sought to determine the techniques that are best supported by the literature [4,12]. Some of the techniques include vascular control, multiple parenchymal transection techniques, various hemostatic agents, low central venous pressure, and hemodilution. These efforts have led to improvements in perioperative outcomes following liver resection. These advances are also associated with the development of hemostatic agents [2]. The progress in hemostatic technology has contributed to improved surgical outcomes and patient survival. The application of THAs to the liver resection surface can also reduce major complications, particularly bleeding and biliary fistulas [13,14].

In liver resection surgery, fibrin glue sealant is commonly used as a THA. However, according to Figueras et al. [15], the application of fibrin sealant to the raw surface of the liver does not appear to be justified. That study found that blood loss, transfusion requirements, the incidence of biliary fistula, and the overall outcomes were comparable to those in patients who did not receive fibrin glue. Therefore, discontinuing routine use of fibrin sealant could result in significant cost savings without compromising patient outcomes.

The effectiveness of MPH, such as Arista^®^ AH, in liver resection compared with other THAs is controversial. To date, no studies have demonstrated the efficacy of MPH in liver resection in humans. However, there are several animal studies that compare the hemostatic capabilities of MPH with those of other THAs. In a study by Lewis et al., hemostatic efficacy was compared using a heparinized porcine abrasion model mimicking a capsular tear of a parenchymal organ. In this study, Arista^®^ AH does not demonstrate superior hemostatic capabilities in hepatic abrasions compared to other hemostatic matrices such as Floseal^®^ [16].

In another animal study investigating severe hepatic hemorrhage in pigs, treatment with novel MPH (Perclot^®^, Baxter) compounds showed significantly improved outcomes compared to standard packing, including higher survival rates, reduced blood loss, and faster application [17,18]. In liver resection, Arista^®^ AH can be especially effective in patients with diffuse bleeding characterized by broad oozing on the liver surface. Additionally, this study highlights that in patients undergoing major resections without cirrhosis, the efficacy of MPH like Arista^®^ AH is more pronounced compared to other groups. Arista^®^ AH’s rapid blood absorption and expansion at the bleeding site concentrate platelets and clotting factors, which is especially advantageous in major resections with large surface areas. Its capability to uniformly cover broad and irregular surfaces ensures thorough coverage of the resection areas, thereby making it extremely effective in managing hemorrhage.

Our study also highlights the efficacy of Arista^®^ AH in significantly reducing postoperative drain output, particularly in patients undergoing major, non-cirrhotic liver resections. This effect is largely attributed to Arista^®^ AH’s unique formulation as a microporous polysaccharide hemosphere. Upon application to the resected liver surface, Arista^®^ AH actively absorbs water and low-weight molecular compounds from the blood, thereby concentrating vital platelets and clotting proteins at the site. This accelerates the natural clotting process, leading to more rapid and effective hemostasis. Consequently, this efficient control of bleeding translates into a reduced accumulation of fluid in the postoperative phase, thereby decreasing the overall volume of drainage required. These findings not only demonstrate Arista^®^ AH’s role in enhancing surgical efficiency but also suggest its potential to improve postoperative patient management by reducing complications associated with excessive fluid accumulation.

In addition, since the components of thrombin-type hemostatic agents that act directly on the hemostatic cascade have bovine or humanized synthesis, there is a risk of complications by inducing a foreign body reaction (FBR) in vivo [16,19]. However, as the powder-type hemostatic product uses plant-based raw materials, the chance of inducing an FBR is reduced, and rapid biodegradation can be expected [20]. In a study comparing OOZFIX^®^ (Theracion Biomedical, Seongnam, Republic of Korea), a new polysaccharide hemostatic agent, with Arista^®^ AH, both products showed comparable hemostatic performance in animal models, with both agents demonstrating minimal foreign body reactions that resolved within two weeks [18]. This study suggests that MPH like Arista^®^ AH is a safe and effective alternative to existing hemostatic agents.

The limitation of this study is the differing recruitment times for the two groups, which might introduce factors affecting the results. While our study maintained consistency in surgical methods, surgeons, and other equipment, thereby isolating the impact of the hemostatic agent, but the different recruitment times for the two groups could potentially influence the outcome. Additionally, due to the study’s emphasis on short-term outcomes, it was not possible to assess long-term outcomes such as patient survival or the potential long-term impacts of using Arista. Therefore, further research is needed, such as large-scale randomized controlled trials, to validate these findings and provide more comprehensive insights into the effects of hemostatic agents like Arista^®^ AH in liver resection procedures.

Our study is the first to prospectively demonstrate the effectiveness of Arista in terms of improving perioperative outcomes, particularly hemostasis, complications, and hospital stay in patients undergoing liver resection.

## 5. Conclusions

Arista^®^ AH significantly improved intraoperative blood management and postoperative recovery in patients undergoing liver resection, particularly in non-cirrhotic patients undergoing major resection. This suggests that the use of Arista^®^ AH in liver resection can positively influence patient outcomes.

## Figures and Tables

**Table 1 medicina-60-00278-t001:** Baseline characteristics of the patients.

	Arista^®^ AH (*n* = 45)	Control (*n* = 156)	*p*-Value
Sex (F/M), *n* (%)	13 (28.9%)/32 (71.1%)	38 (24.4%)/118 (75.6%)	0.462
Age, years, median (IQR)	55.8 (±15.86)	60.24 (±13.72)	0.170
Underlying liver disease, *n* (%)			
HBV	31 (68%)	117 (77.5%)	
HCV	0 (0%)	3 (2.5%)	
Alcohol	3 (6.7)	8 (5.1%)	
NBNC	11 (24.4%)	28 (17.8%)	
AST	48.2 (±30.6))	47.1 (±28.9)	0.175
ALT	48.1 (±29.3)	46.5 (±26.3)	0.262
PT INR	1.1 (±0.1)	1.2 (±0.1)	0.476
Creatinine (mg/dL)	0.9 (±0.5)	0.9 (±0.8)	0.986
BMI, kg/m^2^	23.09 (±4.8)	24.4 (±3.15)	0.804
Tumor Size, cm	3.30 (±1.88)	3.38 (±1.68)	0.683
Cirrhosis, *n* (%)	18 (40%)	61 (39.1%)	0.914
Liver resection, Major/Minor	15/30	46/110	0.713

Abbreviations: ALT, alanine aminotransferase; AST, aspartate aminotransferase; BMI, body mass index; F, female; HBV, hepatitis B viral infection; HCV, hepatitis C viral infection; IQR, interquartile range; M, male; NBNC, non-B, non-C hepatocellular carcinoma; PT INR, prothrombin time–international normalized ratio.

**Table 2 medicina-60-00278-t002:** Subgroup based on resection type and cirrhosis status.

	Arista^®^ AH (*n* = 45)	Control (*n* = 156)
Minor resection, non-cirrhotic, *n* (%)	15 (33.3%)	67 (42.9%)
Minor resection, cirrhotic, *n* (%)	15 (33.3%)	43 (27.6%)
Major resection, non-cirrhotic, *n* (%)	12 (26.7%)	25 (16.0%)
Major resection, cirrhotic, *n* (%)	3 (6.7%)	21 (13.5%)

**Table 3 medicina-60-00278-t003:** Comparison of periprocedural outcomes.

	Arista^®^ AH (*n* = 45)	Control (*n* = 156)	*p*-Value
Operation time, min	155.56 (±67.0)	179.01 (±124.5)	0.099
EBL, mL	495.56 (±672.7)	691.9 (±777.5)	0.049
Blood transfusion, *n* (%)	6 (13.3%)	30 (19.2%)	0.508
Drain fluid collected within 48 h, mL	337.13 (±273.8)	441.13 (±388.3)	0.045
Time to drain removal, days	4.64 (±1.31)	5.30 (±2.87)	0.030
Duration of hospital stay, days	5.93 (±1.88)	6.94 (±4.17)	0.024
Overall complications, *n* (%)	10 (22.2%)	39 (25%)	0.853
Bleeding related complications	4 (8.9%)	12 (7.7%)	
Bile leakage	1 (1.1%)	4 (2.6%)	
Clinically relevant complication, *n* (%)	3 (6.7%)	12 (7.7%)	0.888

Abbreviation: EBL, estimated blood loss.

**Table 4 medicina-60-00278-t004:** Comparison of periprocedural outcomes and recurrence patterns.

Minor Resection, Non-Cirrhotic	Arista^®^ AH (*n* = 15)	Control (*n* = 67)	*p*-Value
EBL, mL	173.3 (±136.1)	270.1 (±177.3)	0.051
Blood transfusion, *n* (%)	1 (6.7%)	0 (0%)	0.183
Drain fluid collected within 48 h, mL	118.6 (±52.5)	180.5 (±42.9)	0.221
Time to drain removal, days	3.79 (±0.80)	3.77 (±0.79)	0.932
Duration of hospital stay, days	5.0 (±1.69)	4.93 (±1.54)	0.868
**Minor resection, cirrhotic**	**Arista^®^ AH** **(*n* = 15)**	**Control (*n* = 43)**	***p*-Value**
EBL, mL	516.67 (±1035)	605.8 (±1562.1)	0.953
Blood transfusion, *n* (%)	3 (20%)	12 (27.9%)	0.736
Drain fluid collected within 48 h, mL	302.5 (±164.2)	351.1 (±142.6)	0.279
Time to drain removal, days	4.73 (±1.58)	5.40 (±2.20)	0.283
Duration of hospital stay, days	6.13 (±2.23)	6.84 (±2.80)	0.383
**Major resection, non-cirrhotic**	**Arista^®^ AH** **(*n* = 12)**	**Control (*n* = 25)**	***p*-Value**
EBL, mL	780 (±400.8)	1076 (±777.5)	0.036
Blood transfusion, *n* (%)	1 (8.3%)	9 (36.0%)	0.076
Drain fluid collected within 48 h, mL	556.1 (±288.7)	735.4 (±333.5)	0.049
Time to drain removal, days	5.08 (±0.8)	6.24 (±2.04)	0.019
Duration of hospital stay, days	6.33 (±0.98)	7.56 (±2.13)	0.013
**Major resection, cirrhotic**	**Arista^®^ AH** **(*n* = 3)**	**Control (*n* = 21)**	***p*-Value**
EBL, mL	600 (±173)	1852.4 (±1336.9)	0.025
Blood transfusion, *n* (%)	1 (33.3%)	9 (42.9%)	0.754
Drain fluid collected within 48 h, mL	727 (±316.8)	987.6(±288.3)	0.290
Time to drain removal, days	6.33 (±1.16)	7.33 (±1.53)	0.291
Duration of hospital stay, days	8.00 (±2.0)	10.43 (±4.52)	0.448

Abbreviation: EBL, estimated blood loss.

## Data Availability

The data presented in this study are available in insert article.

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
