# Peer review of "A Prospective Analysis of the Effects of a Powder-Type Hemostatic Agent on the Short-Term Outcomes after Liver Resection"

_medicina, 2024, doi:10.3390/medicina60020278_

Round 1
Reviewer 1 Report
Comments and Suggestions for Authors
Dear authors,
I am delighted to have been invited to review your interesting paper on the use of Arista to control post-liver resection bleeding. Overall, I found it well-written, engaging, easy to follow and well-structured, with no need for major changes or English language review.
A couple of points that I would like to highlight and might need addressing are the following:
1. Arista was mostly used in minor resections (30/45) and despite the fact that there was no statistically significant difference, the number of events you are describing (15 major resections, out of which only three were cirrhotic) might actually have an impact (number of events).
2. I would appreciate a comment on liver quality (degree of steatosis, impact on outcomes?) if these data are available, as well as some comorbidities of these patients (ASA scores, DM, CKD, CVD).
3. I noticed that tables 3 and 4 showcase a benefit for Arista in EBL mostly in the non-cirrhotic group (as there was no diff in the minor cirrhotic and I feel that drawing statistical conclusions with only 3 major resections in the cirrhotic group might not translate clinically).
4. "The results show that Arista® AH significantly improved the perioperative outcomes after liver resection." - moderate rephrasing, adding generally.
5. Do you feel that the 3rd paragraph of the discussion would fit better in the introduction?
6. I would appreciate a clear statement of the strengths and limitations of your study, taking all the above into consideration and discussing the clinical versus statistical relevance of your study as well as the need for more robust testing in major resections in cirrhotic patients.
Kind regards.
Author Response
"Please see the attachment."

Reviewer 2 Report
Comments and Suggestions for Authors
This manuscript titled "Prospective Analysis of the Effects of a Powder-Type Hemostatic Agent on Short-Term Outcomes After Liver Resection." is an analysis of liver resections using Arista vs. standard hemostatic agents. It is well written and Arista appears to be safe to use in this setting. I have some comments:
1. The sample size is small and makes this difficult to interpret. Even further the use of subgroup analysis has very small numbers. Conclusions were made from these comparisons and this cannot be done based on this data.
2. The use of Mean EBL is difficult to interpret. One outlier can skew small sample sizes creating a difficult to interpret result.
3. The drain fluid outcome is also challenging to interpret. No mention of drain consistency is noted. In cirrhotic patients who have more liver resected, ascites may occur which has nothing to due to the hemostatic agent used.
4. Arista was applied after most bleeding controlled I would imagine. Not sure EBL is really a useful thing to measure if we are worried about post operative outcome. Most agents are applied at the end of a case prior to closure which would lead me to review post op transfusion. Intraoperative findings are less useful with regards to the intervention at hand.
5. No mention of cost was discussed and this should be mentioned especially if repeated application was required.
Author Response
"Please see the attachment."

Reviewer 3 Report
Comments and Suggestions for Authors
1. The last 4th paragraph in the Discussion section, it is better to tell readers what's difference between Arista PMA P050038 and Arista AH powder?
2. If possible, please add more about the effectiveness of other THA in the discussion section.
Author Response
"Please see the attachment."
